# Protein C Pretreatment Protects Endothelial Cells from SARS-CoV-2-Induced Activation

**DOI:** 10.3390/v16071049

**Published:** 2024-06-28

**Authors:** Bruna Rafaela dos Santos Silva, Davi Sidarta-Oliveira, Joseane Morari, Bruna Bombassaro, Carlos Poblete Jara, Camila Lopes Simeoni, Pierina Lorencini Parise, José Luiz Proenca-Modena, Licio A. Velloso, William H. Velander, Eliana P. Araújo

**Affiliations:** 1School of Nursing, Universidade Estadual de Campinas, Campinas 13083-970, SP, Brazil; 2Laboratory of Cell Signalling, Obesity and Comorbidities Center (OCRC), Universidade Estadual de Campinas, Campinas 13083-970, SP, Brazil; 3School of Medical Sciences, Universidade Estadual de Campinas, Campinas 13083-970, SP, Brazil; 4Surgery Department, Omaha VA Medical Center, Omaha, NE 68105, USA; 5Laboratory of Emerging Viruses, Institute of Biology, Universidade Estadual de Campinas, Campinas 13083-970, SP, Brazil; simeonicamila@gmail.com (C.L.S.); pierinalp@gmail.com (P.L.P.); jlmodena@unicamp.br (J.L.P.-M.); 6Department of Chemical and Biomolecular Engineering, University of Nebraska, Lincoln, NE 68588, USA; wvelander2@unl.edu

**Keywords:** bioinformatics, blood coagulation disorders, endothelial cell, SARS-CoV-2, inflammation

## Abstract

SARS-CoV-2 can induce vascular dysfunction and thrombotic events in patients with severe COVID-19; however, the cellular and molecular mechanisms behind these effects remain largely unknown. In this study, we used a combination of experimental and in silico approaches to investigate the role of PC in vascular and thrombotic events in COVID-19. Single-cell RNA-sequencing data from patients with COVID-19 and healthy subjects were obtained from the publicly available Gene Expression Omnibus (GEO) repository. In addition, HUVECs were treated with inactive protein C before exposure to SARS-CoV-2 infection or a severe COVID-19 serum. An RT-qPCR array containing 84 related genes was used, and the candidate genes obtained were evaluated. Activated protein C levels were measured using an ELISA kit. We identified at the single-cell level the expression of several pro-inflammatory and pro-coagulation genes in endothelial cells from the patients with COVID-19. Furthermore, we demonstrated that exposure to SARS-CoV-2 promoted transcriptional changes in HUVECs that were partly reversed by the activated protein C pretreatment. We also observed that the serum of severe COVID-19 had a significant amount of activated protein C that could protect endothelial cells from serum-induced activation. In conclusion, activated protein C protects endothelial cells from pro-inflammatory and pro-coagulant effects during exposure to the SARS-CoV-2 virus.

## 1. Introduction

Coagulopathy and cardiovascular complications are the most important outcomes of severe coronavirus disease 19 (COVID-19) [1,2]. Studies have shown that endothelial cell (EC) dysfunction is central to this process [3,4]. These data suggest that reduced vascular integrity in COVID-19-positive patients could expose prothrombotic subendothelial factors, resulting in platelet capture, activation of coagulation cascades, thrombin activation, and fibrin production [5]. To support this, there is an EC-dependent inflammatory response that intensifies the hypercoagulability phenomenon and potentially results in disseminated intravascular coagulation [6].

Healthy blood vessels are lined by a monolayer of ECs that plays a crucial role in preventing the formation of pathological thrombosis. This is due, at least in part, to the presence of several receptors expressed on ECs that act to promote anticoagulant pathways [7]. The activated protein C (aPC) is a serine protease derived from its inactive zymogen, protein C (PC). PC activation is optimally performed on the EC surface when PC binds to its receptor, the endothelial protein C receptor (EPCR), through its γ-carboxyglutamic acid (Gla) domain, and the activator thrombin is linked to EC-thrombomodulin (TM) [8]. Endothelial cells play an important role in health hemostasis [9,10]. Endothelial derangement can occur in several pathological conditions, such as infections and autoimmune diseases, contributing to the coagulopathy that frequently accompanies these conditions [11]. The severe acute respiratory syndrome coronavirus 2 (SARS-CoV-2) mainly infects the respiratory tract and usually progresses rapidly to acute respiratory distress syndrome (ARDS), with subsequent systemic viral spread and cytokine storm to cause endotheliitis of the vasculature of the intestine, kidney, heart, and brain [12].

Thrombotic events related to hypercoagulability were widely demonstrated in studies conducted on lung autopsies of patients affected by severe COVID-19, in which they demonstrated diffuse alveolar damage in the acute phase with the formation of hyaline membranes with exudate rich in fibrin in the alveolar space, which could hinder the gas exchange [13]; in addition; there was alveolar destruction with bloody exudate, dilation of capillaries with congestion, mononuclear infiltration, and thrombosis [14]. Lung and cardiac autopsies revealed edematous and rigid lung parenchyma with focal hemorrhage, while cardiomegaly and dilation of the right ventricle were observed in the heart, often associated with elevated brain natriuretic peptide (BNP). Microscopic findings included an inflammatory infiltrate with the predominance of CD4+ and CD8+ T lymphocytes and pulmonary megakaryocytes within the alveolar capillaries with signs of active platelet production, and additionally, arterial and venous thrombi were found [15].

Fogarty and collaborators [16] described that coagulopathy was the main complication in a cohort of 67 patients with COVID-19, reflecting the crucial role of ECs in this disease and suggesting that advances in understanding its pathophysiology could lead to the identification of new potential therapeutic targets. However, despite tremendous advances in the field, the cellular and molecular mechanisms by which SARS-CoV-2 infection triggers coagulopathy remain elusive. Histological analyses of COVID-19 patients and in vitro experiments demonstrated that ECs do not support the productive replication of SARS-CoV-2. Still, direct exposure to SARS-CoV-2 induces the secretion of pro-inflammatory cytokines and adhesion molecules [17,18]. This supports the hypothesis that virus components could mediate the endothelial dysfunction observed in COVID-19 patients.

The PROWESS clinical trial demonstrated that activated recombinant human protein C (rhAPC), compared with placebo, reduced the incidence of deaths in adults with severe sepsis and high risk of death (APACHE II > 25), but with a higher risk of bleeding [19]. These results culminated with approval by the Food and Drug Administration (FDA), USA, in 2001 for the use of rhAPC (Xigris, Eli Lilly, Indianapolis, IN, USA) in patients with severe sepsis. After approval, several studies were carried out to confirm the efficacy and safety of rhAPC, but without success. According to the clinical study PROWESS-SHOCK [20], which did not prove the effectiveness of rhAPC in sepsis, the manufacturer decided to withdraw the drug from the market. However, other drugs that modulate the PC pathway have been studied in clinical trials in the expectation of a therapeutic role for rhAPC based on laboratory and experimental studies on the antithrombotic, pro-fibrinolytic, anti-inflammatory, and cytoprotective properties of ECs and, in theory, they are highly beneficial during inflammatory disease states, many of which depend on signaling through EPCR and PAR-1 [21,22].

In patients with severe sepsis, it is common to observe a decrease in serum aPC levels with higher baseline aPC levels in survivors [23]. In this sense, new uses for PC deserve to be explored [24]. Despite the inconclusive outcome of rhAPC in sepsis, the characteristics observed in patients with severe COVID-19 have mobilized the scientific community to focus on aPC, with potential benefits of the management of thromboembolic complications in these patients [22,25].

Finally, the COMPASS-COVID-19 trial designed to validate predictors for the worsening of COVID-19 in patients admitted to the intensive care unit (ICU) described the decrease in protein C as a predictive risk factor for progression to COVID-19. In these patients, antithrombin and protein C activity levels were lower, while D-dimer and fibrinogen levels were higher compared with patients admitted to a conventional unit [26].

Here, we hypothesize that PC could play a protective role against SARS-CoV2-induced EC coagulopathy and inflammatory response. To test this hypothesis, we used bioinformatics to access vascular ECs through single-cell transcriptomes from healthy controls and COVID-19 patients (Figure 1A). Additionally, primary human umbilical vein endothelial cells (HUVEC) were exposed to SARS-CoV-2 particles or incubated with serum from patients with severe COVID-19 (Figure 1B). Then, we explored PC signaling components and endothelial activation genes in the treated cells. We identified changes in the expression of pro-inflammatory, pro-coagulation, and cytoprotective genes in ECs exposed to viral infection that were partly reversed by the PC treatment. This study suggests the PC system may contribute to restoring the local and systemic hemostatic derangement in COVID-19.

## 2. Materials and Methods

Cell culture. Primary human umbilical vein endothelial cells (HUVECs) were seeded in 12-well plates and incubated in 45% RPMI (Invitrogen, Waltham, MA, USA), 45% 199 (Sigma-Aldrich, Burlington, VT, USA) medium with 10% fetal bovine serum (FBS) at 37 °C in a 5% CO_2_ incubator. Experiments were performed when cells reached approximately 90% confluence per well.

Protein C pre-treatment. The dose and concentration of protein C (IHUPROCAI100UG, Innovative Research) used in the experiments were based on serum PC levels in human plasma [27] and standardized according to the dose test and treatment time used previously. Cells at a confluence of 90% were washed with PBS before PC treatment, and the medium (50% RPMI, 50% 199) without FBS was used for incubation. Approximately 400 μL of medium with purified human inactive protein C (0.8 ng/μL) was added to the wells for 4 h. After incubation, the medium was removed, and exposure to SARS-CoV-2 Delta variant occurred, or patients’ sera were added.

Viral infection. HUVECs of different passages were seeded in 12-well plates and incubated in 45% RPMI, 45% 199 medium with 10% FBS at 37 °C under 5% CO_2_. This set of experiments was performed in four distinct groups. Control group (CLT): treated only with culture medium without FBS. SARS-CoV-2 group: cells were exposed to infection by the Delta variant of SARS-CoV-2. Protein C group (PTPC): cells were pre-treated for 4 h with purified human inactive protein C (0.8 ng/μL). PTPC + SARS-CoV-2 group: cells were pre-treated for 4 h with purified human inactive protein C (0.8 ng/μL) and then exposed to infection by the Delta variant of SARS-CoV-2. The SARS-CoV-2 Delta variant used in this study was kindly provided by the Emerging Viruses Laboratory of the University of Campinas, Brazil. Cells were infected with 4 × 10^5^ FFU/mL of virus and an MOI of 0.1 for 1 h at 37 °C, 5% CO_2_. The viral inoculum was then removed, and the medium was replaced with 50% RPMI and 50% 199 medium without FBS. After 24 h, medium and cell lysates were collected for ELISA testing and quantitative real-time PCR (RT-qPCR), respectively. All experiments with SARS-CoV-2 were performed under physical containment 3 (PC3) conditions.

Incubation of HUVECs with serum from healthy subjects or severe COVID-19 patients. A serum sample from a patient with severe COVID-19 collected upon admission to the Intensive Care Unit (ICU) and from a healthy subject (control group) collected before the COVID-19 outbreak were used at a concentration of 5% to incubate HUVECs for 4 h. The HUVECs were seeded as previously described, and experiments were performed when cells reached a confluence of approximately 90% per well. This set of experiments was performed with six groups. Control group (CTL): treated only with culture medium (50% RPMI and 50% 199). PTPC group: cells were pre-treated for 4 h with purified human inactive protein C (0.8 ng/μL). Severe serum group: cells were incubated for 4 h with culture medium at a concentration of 5% of serum from a patient with severe COVID-19. PTPC + Severe serum group: cells were pre-treated with purified human inactive protein C at 0.8 ng/μL for 4 h; after incubation, protein C was removed, and then a culture medium with 5% serum from a severe COVID-19 patient was added for 4 h. Healthy serum group: cells were treated for 4 h with a culture medium at a concentration of 5% of serum from a healthy patient. PTPC + Healthy serum group: cells were pre-treated for 4 h with purified human inactive protein C at 0.8 ng/μL; after incubation, the protein C was removed, and culture medium at a concentration of 5% of healthy serum was added for 4 h. After incubation, the cell lysate was collected for subsequent RNA extraction and analysis by quantitative real-time PCR (RT-qPCR). All studies with human serum were previously approved by the Research Ethics Committee, number 62864822.8.0000.5404.

RNA extraction and cDNA synthesis. Total RNA was isolated from HUVECs using Trizol reagent according to the manufacturer’s recommendations (Life Technologies, Frederick, MD, USA). cDNA was synthesized using the high-capacity cDNA reverse transcription kit (Life Technologies) according to the manufacturer’s guidelines.

RT-qPCR Array. A total of 20 ng of reverse-transcribed cDNA from HUVEC viral infection and protein C treatment were used in each reaction of a real-time RT-qPCR array (RT² Profiler™ PCR Array Human Protease Activated Receptor Signaling PAHS-159Z, Qiagen, Redwood City, CA, USA) containing 84 related genes. Analyses were performed at https://geneglobe.qiagen.com/us/analyze (accessed on 16 March 2023). Modulated genes were selected as candidate genes and validated by RT-qPCR (Appendix A).

Quantitative PCR (RT-qPCR). RT-qPCR was performed using LuminoCT PCR Master Mix (Sigma-Aldrich). The candidate genes obtained in the RT-qPCR Array and other genes of interest were evaluated. A total of 20 ng of reverse-transcribed cDNA were used in each reaction. The assay code of the primers is listed in Appendix A. Each PCR contained 20 ng of cDNA, 3 μL of LuminoCT PCR master mix (Sigma-Aldrich^®^, Burlington, VT, USA), 0.25 μL of primers, and 0.25 μL of ultrapure water and was analyzed in the QuantStudio 6 real-time PCR system (Applied Biosystems, Foster City, CA, USA). Each sample was tested in duplicate, and data were normalized to the average of two housekeeping genes (*PPIA* and *GAPDH*) and expressed as relative mRNA levels using the 2^−ΔCt^ method.

Viral quantification by RT-qPCR. The quantification of *SARS-CoV-2* was performed by one-step RT-qPCR using the set of primers: Forward: 5-ACA GGT ACG TTA ATA GTT AAT AGC GT-3; Reverse: 5-ATA TTG CAG CAG TAC GCA TAC GCA CAC A-3; Probe: 5–6FAM-ACA CTA GCC ATC CTT ACT GCG CTT CG-QSY-3. Each PCR contained 3 μL (20 ng) of total RNA, 4 μL of qPCRBIO Probe 1-Step Go (PCRBIOSYSTEMS, London, UK), 1.0 μL of each primer, 0.5 μL of the probe, and 0.5 μL of RTaseGo (PCRBIOSYSTEMS, London, UK). No infection was observed in the HUVEC cultures, as the final concentration of the virus in the cells was approximately 2 × 10^3^ FFU/mL (Appendix A).

Enzyme-linked immunosorbent assay—ELISA. Activated Protein C levels were measured using the human activated protein C (Sandwich ELISA) ELISA Kit LS-F54930 (LSBio, Shirley, MA, USA). The assay was performed following the manufacturer’s protocol. Detection was performed on the microplate reader (MicroplateReader BioTek 800 Elx, Agilent Technologies, Santa Clara, CA, USA) and measurement was immediately conducted at 450 nm.

Single-cell transcriptomics analyses. Datasets were retrieved from the Gene Expression Omnibus (GEO) and the European Genome-phenome Archive (EGA) under accession numbers GSE171668 and EGAS00001004344, respectively. The raw data with the .h5 format were processed using the standard toolkit Scanpy 1.7.2 in Python igraph 0.9.6. Processing and quality control flow have been previously described [28] and are available at https://scanpy-tutorials.readthedocs.io/en/latest/pbmc3k.html (accessed on 5 July 2023). Data were then integrated using Harmony’s algorithm, which designs cells in a shared embedding grouped by cell type [29]. Cell-type identity was manually annotated, and differentially expressed genes (DEG) for each cluster and remaining cells were calculated using Wilcoxon’s two-sided rank sum test [30].

Statistical analysis. The differentially expressed genes (DEGs) between endothelial cells of patients with COVID-19 compared with healthy controls were determined using a two-sided Wilcoxon rank-sum test Scanpy [30]. The *p*_val_cutoff = 0.05 and logfc_cutoff = 0.1 were established by the Scanpy utils function. GraphPad Prism version 8.0 was used for statistical analysis of real-time PCR gene expression and ELISA assays. Data are presented as mean ± s.e.m. with individual data points indicated. One- and two-way ANOVA followed by Tukey’s multiple-comparison tests were used for statistical analyses. * *p* ≤ 0.05, ** *p* < 0.01, *** *p* < 0.001, and **** *p* < 0.0001 compared with the indicated groups.

Generation of Diagrams and Illustrations. A licensed version of BioRender (Web online version 2024. Accessed on 19 June 2024) was employed to generate the diagrams present in most figures.

## 3. Results

### 3.1. Expression of Genes Involved in Endothelial Damage and the PC Pathway in Human Lung Endothelial Single Cells

Although COVID-19 is primarily a respiratory disease, recent studies suggest that endothelial dysfunction may exacerbate deleterious events by inciting thrombotic, inflammatory, and microvascular processes, thus contributing to its extrapulmonary complications [12]. To elucidate the mechanisms that drive endothelial damage in patients with COVID-19, we compared the expression of genes involved in endothelial activation and activated protein C (aPC) signaling in pulmonary ECs from patients who died from COVID-19 and compared with lung ECs from healthy subjects in publicly available datasets. As a result, we identified gene encoding receptors involved in aPC signaling; F2R, S1PR1, and NOS3 were predominantly expressed by arterial and venous cells in the healthy lung. In addition, S1PR1 was expressed by most endothelial subtypes (Figure 2A). The expression of pro-inflammatory and pro-coagulation-related genes ITGAM, PLEK, and vWF was detected by most cell subtypes (Figure 2A). In ECs obtained from deceased patients who had COVID-19, the arterial subtype also predominantly expressed F2R, S1PR1, and NOS3. However, unlike in the healthy lung, S1PR1 was less expressed in the other subtypes (Figure 2B,C). Additionally, we identified a decrease in the expression of F2R (*p* < 0.001), S1PR1 (*p* = 0), and NOS3 (*p* < 0.001) compared with healthy ECs (Table 1; Figure 2C). However, in patients with COVID-19, the expression of ITGAM, vWF, and PLEK was higher when compared with ECs of healthy patients (Table 1; Figure 2C).

### 3.2. PC Pretreatment Reduces the Expression of CCL2, IL6, and SERPINE1 Genes in HUVECs Exposed to SARS-CoV-2 Infection

Previous studies suggest that the endothelial dysfunction seen in severe COVID-19 is possibly due to viral and pro-inflammatory proteins released by adjacent cells infected with SARS-CoV-2 rather than direct virus infection [17,18]. To explore EC dysfunction in the context of COVID-19, we exposed HUVECs to infection by the Delta variant of SARS-CoV-2. We did not find active viral replication in the cell cultures (Figure 3A). Nevertheless, as our previous study suggested, protein C could play an important role in hypercoagulability in severe COVID-19 [28]. We asked whether treatment of ECs with PC before exposure to SARS-CoV-2 infection could result in a protective endothelial effect. For this, HUVECs were pretreated for four hours with purified inactive human protein C (IHUPROCAI100UG, Innovative Research) and subsequently exposed to infection by the Delta variant of SARS-CoV-2. In cells that received pre-treatment with PC and then were exposed to SARS-CoV-2 infection, the expression of pro-inflammatory and pro-coagulant genes CCL2, IL6, and SERPINE1 significantly decreased compared with untreated cells (Figure 3B–D). Furthermore, CAV1 and PROC expression were also decreased in cells pretreated with PC compared with cells exposed to SARS-CoV-2 (Figure 3E–G). There was no significant modulation in the gene expression of vWF, VCAM1, F3, SELE, ITGAM, F2R, S1PR3, NOS3, PTGS2, PLEK, RHOH, TNF-a, and IL-3 between the groups (Appendix A).

### 3.3. SARS-CoV-2 Does Not Promote Protein C activation in HUVECs

PC activation occurs through binding to the thrombin-thrombomodulin complex on the surface of endothelial cells; which is enhanced by the EPCR [8]. Here, we used inactive human protein C as pretreatment and hypothesized whether exposure to SARS-CoV-2 could promote its activation. Cells that received PC pretreatment with or without viral exposure showed no significant differences in the amounts of aPC (Figure 3H).

### 3.4. Activated PC Present in Severe COVID-19 Serum May Protect ECs from Increased Expression of Inflammatory and Pro-Coagulation Genes

To mimic severe COVID-19, HUVECs were incubated for 4 h in the serum of a patient with critical COVID-19; also, ECs were pretreated or not with PC before serum exposure. When incubating HUVECs with serum from a healthy individual, we observed a significant decrease in the expression of CCL2 and IL6 compared with the control pretreated with PC (Figure 4A,B). Surprisingly, HUVECs incubated with severe COVID-19 serum also showed a decrease in the expression of CCL2, IL6, TNFa, PTGS2, and ITGAM compared with the control that received or did not receive pretreatment with PC (Figure 4A–E). Compared with serum from a healthy individual, we observed a decrease in the expression of TNFa and F3 in cells incubated with severe COVID-19 serum (Figure 4C,F). Pretreatment with PC before incubation with severe COVID-19 serum did not significantly alter the expression of inflammatory markers (Figure 4A–G). No significant differences in the gene expression of SERPINE-1, SELE, PLEK, CAV-1, PROC, NOS-3, KDR, and S1PR-3 were observed between the groups (Appendix A). To understand the mechanism behind the effect of severe COVID-19 serum on HUVECs, aPC amounts were determined by ELISA (Figure 4H–J). Severe serum showed a trend to increase aPC compared with healthy serum. When PC was added before the incubation with healthy serum, the amount of aPC decreased significantly compared with healthy serum without PC pretreatment (probably owing to the dilution of the aPC) (Figure 4H). When a pretreatment with PC before incubation with severe serum occurred, the same effect was observed (without statistics), but the proportion of a higher amount of aPC in the severe COVID-19 serum remained (*p* = 0.09) (Figure 4H). When adding inactive PC to severe and healthy serum, a small amount of aPC was detected, meaning that PC did not become aPC (Figure 4I,J).

## 4. Discussion

Solid evidence shows that the vascular endothelium plays a key role in severe COVID-19 [12,31,32]. However, the mechanisms underlying this dysfunction remain mostly unclear. In this work, we used single-cell RNA sequencing transcriptomic data to access EC populations from patients who died as a result of severe COVID-19 and compare them with data from healthy subjects. Furthermore, we exposed HUVECs to SARS-CoV-2 infection and incubated these cells with serum from patients with severe COVID-19. We showed ECs were not infected with the virus; however, they underwent transcriptional alterations of inflammatory, coagulation, and cytoprotective genes. Some of the transcriptional changes observed in virus-exposed cells were reversed in cells pretreated with PC, demonstrating a modulating effect that extends beyond immunological changes to changes in coagulation and endothelial barrier stability, which are essential for hemostasis. Additionally, we observed that serum from a patient with severe COVID-19 contains more aPC than serum from healthy subjects, suggesting a protective effect of ECs from serum-induced activation. Thus, endothelial damage from COVID-19 could be, at least in part, mitigated by the aPC pathway.

The capacity or not of SARS-CoV-2 to directly infect ECs is a question with no consensus in the literature [17,18,33], since angiotensin-converting enzyme (ACE) 2 expression is relatively low in these cell types [17]. Static model study suggests that, even if SARS-CoV-2 could infect ECs, they are not capable of sustaining an active replication [17]. Conversely, the 3D microphysiological system platform model recently demonstrated that the expression of ACE2 in ECs is stimulated by shear stress, and in this context, ECs are not only more susceptible to viral infection but also secrete pro-inflammatory cytokines, including IL-6 along with coagulation factors to activate downstream, non-infected ECs, providing an amplification mechanism for inflammation and coagulopathy [34].

Furthermore, several studies have indicated that the significant endothelial dysfunction observed in COVID-19 is due to autoantibodies [35] and viral and pro-inflammatory proteins released by adjacent cells infected with SARS-CoV-2 [36,37]. In our study, we also did not observe infection or sustained viral replication in the HUVEC cultures studied. The susceptibility of ECs to the virus was evaluated by Urata and collaborators [38], who found that SARS-CoV-2 infects HUVECs; however, it does not replicate and disappears within 72 h without causing serious cell damage. Furthermore, HUVECs with earlier passage are less susceptible to SARS-CoV-2 infection than senescent cells. Finally, more gene expression is affected in senescent ECs by SARS-CoV-2 infection than in early passage ECs. Here, although *VWF*, *VCAM1*, *F3*, *ITGAM*, *S1PR3*, *PTGS2*, *PLEK*, and *RHOA* markers have no statistical significance-sustained increase compared with controls, it should be noted that exposure to the virus showed a trend of increase in their expression (Figure 3 and Appendix A). These changes could be more pronounced in senescent HUVECs, justifying the greater severity of COVID-19 in the elderly.

In addition to the increase in inflammatory and coagulation proteins, endothelial dysfunction in COVID-19 is related to the downregulation of cytoprotective proteins [18,39]. The aPC pathway is initiated by the formation of the complex of thrombin, thrombomodulin, and the endothelial protein C receptor (EPCR), allowing the conversion of vitamin K-dependent zymogenic circulating PC into its activated form (aPC) on the cell surface [40]. Initial studies of gene expression profiling in HUVECs implicated broader biological activities of aPC. Administration of aPC to HUVECs after stimulation with TNF-α resulted in anti-apoptotic signals that promote cell survival [41]. aPC also reduces the endothelial expression of chemokine and adhesion molecules in the HUVECs [42]. Human coronary artery ECs exposed to a cocktail of cytokines (TNF-α, IL-1β, IFN-γ) and recombinant aPC show no effect on the induction of endothelial nitric oxide synthase but drastically reduce intercellular adhesion molecule expression-1, as well as IL-6, IL-8 transcript, and monocyte chemoattractant protein-1 (MCP-1) [43]. Despite this, the benefits of the activated form of PC in pathological conditions remain unclear in the literature, where the path for more studies is open.

To control the immunological derangements observed in critically ill patients during the SARS-COV-2 pandemic, the Food and Drug Administration (FDA) approved immunomodulatory drugs. One of them was tocilizumab, a humanized monoclonal antibody that acts by blocking interleukin 6 receptors, for use in certain adults hospitalized with COVID-19 [44]. However, later REMDACTA and EMPACTA studies found that tocilizumab did not reduce all-cause mortality [45,46]. The lack of success of the therapeutic approach may be due to an isolated attempt to manage the immunological disorder or the virus itself rather than the coagulation system. We found that pretreatment of endothelial cells with PC decreased the expression of the immunomodulators *IL6* and *CCL2*, as well as the coagulation marker *SERPINE1*.

The rationale for using the inactive form of PC in this study is primarily based on the short half-life of aPC, which is an average of 20 min compared with 10 h for PC [47]. Moreover, we hypothesize that the aggressiveness of inflammatory proteins and viral exposure is faster and overcomes the endogenous activation capacity of the circulating PC zymogen that may be ineffective in protecting the vascular bed from SARS-CoV-2-induced dysfunction. The inactive PC may be stored in the EC (completing the PC reserve), and as soon as the cell is exposed to the virus, this protein quickly becomes activated and protects the vascular bed (Figure 5).

Therefore, we observed here that PC pretreatment induced a significant decrease in inflammatory and procoagulant genes in cells exposed to SARS-CoV-2 infection compared with cells that did not receive pretreatment (Figure 3). Together, we propose a hypothesis that could explain the findings. First, the protective effect observed in HUVECs exposed to SARS-CoV-2 could be mediated by inactive PC, since the ELISA carried out in the collected culture medium did not indicate activation of the protein (Figure 3H). It is already known that aPC generation requires thrombin [48], and hence the relative lack of it may contribute to the lack of PC activation. The second hypothesis is that experimental evidence demonstrates that the endothelial cell PC receptor (EPCR) can undergo translocation from the plasma membrane to the nucleus, where it redirects gene expression. During translocation, it can transport aPC to the nucleus, possibly being responsible for aPC’s ability to modulate inflammatory mediator responses in the endothelium [49]. It is possible that the inactive PC added was activated and translocated to the cell nucleus, which is why it was not captured in the ELISA test.

The demonstration that aPC is a normal plasma component [50], whose enzymatic activity can be detected with specific and sensitive methods [27,51], indicates that the PC anticoagulant pathway is continuously activated in vivo. In this sense, a long-standing effort has been made to clarify the serum levels of PC and aPC that could be related to thrombotic states, since congenital or acquired disorders characterized by the production or impaired function of aPC are associated with a high risk of venous thromboembolism (VTE) [52]. Clinical and experimental data on PC deficiency support the hypothesis that an aPC deficiency, whether due to impaired PC activation, PC zymogen deficiency, or increased aPC inhibition, can result in a prethrombotic state [52]. However, establishing cut-off serum levels of PC and aPC is still a challenge, since several conditions influence their serum levels. Plasma levels of aPC are commonly increased under pathological conditions. Cattaneo and collaborators [53] described that mean plasma levels of aPC, and aPC/PC ratios were higher in patients with VTE than in healthy controls. Moreover, endogenous aPC formation was increased during endotoxemia [54], and meningococcal sepsis in children [55]. The normal average plasma concentration of PC is equal to 67 nmol/L (4.3 μg/mL), and that of aPC is equal to 38 pmol/L [27]. Two related individuals with semi-normal levels of PC zymogen have semi-normal levels of circulating aPC and suggest that the level of aPC under basal conditions in the absence of hemostatic stress may be proportional to the circulating PC level [27], where physiological aPC/PC ratio is close to 1. However, under pathological conditions, the aPC/PC ratio may increase to protect the host.

Ilmakunnas and collaborators [56], when evaluating preoperative serum levels of PC and aPC in patients who underwent liver transplantation, identified that despite PC deficiency, patients with liver failure managed to maintain normal aPC levels. Protein C levels were found to decrease during surgery, while aPC levels increased. As a result, the aPC/PC ratio showed a significant increase during surgery, reflecting increased aPC formation despite low PC levels. Our previous in silico data [28] revealed a decrease in hepatic expression of *PROC* in patients who died as a result of severe COVID-19. Additionally, previous studies have shown decreased plasma PC levels in patients with COVID-19 at hospital admission [57]; however, here we observed serum levels of aPC relatively higher in serum with severe COVID-19 than in healthy serum (Figure 4H). These data suggest that during PC deficiency in COVID-19, PC activation can increase to preserve blood coagulation homeostasis. Protection is evidenced in our results by the lower induction in the expression of pro-inflammatory and pro-coagulation genes observed in HUVECs incubated with severe COVID-19 serum compared with healthy serum and control.

During sepsis, endogenous aPC production initially increases in apparent response to overcoming an intense procoagulant state. However, overwhelming inflammation ultimately results in deficiency of the PC system with reduced levels of precursor and activated form of PC [58]. A similar phenomenon may occur in severe COVID-19. Subcoagulant amounts of thrombin in the circulation can increase plasma levels of endogenous aPC, which can, therefore, be considered markers of a hypercoagulable state [27]. In the study by Kleijn and colleagues [55], in children with meningococcal sepsis, aPC levels paralleled thrombin markers and were positively related to severity, with the highest levels in non-survivors. Moreover, the serum evaluated in this study was obtained on the first day of admission to the ICU, and the outcome of this patient was death.

It is inevitable to note that severe COVID-19 serum has induced lower inflammatory and pro-coagulation gene expression in HUVECs compared with healthy serum and culture medium (Figure 4A–D). These results oppose the endothelial activation induced by serum from COVID-19 patients described in the literature [35]. However, other studies observed that incubation with serum from COVID-19 patients altered the surface marker expression profile in HUVEC but did not induce an EC-activated phenotype. Instead, post-COVID-19 syndrome (PCS) serum and PCS with chronic fatigue syndrome (CFS) serum led to a significant reduction in surface expression of VCAM-1 and E-selectin compared with healthy control serum samples [59]. Vieceli and collaborators [60] also evaluated transcriptional changes, permeability, and thrombin generation in HUVECs incubated with serum from COVID-19 patients. They observed an increased expression of F3 (Tissue Factor [TF]) and decreased EPCR; however, other molecules involved in the endothelial regulation of coagulation, such as PLAT (tPA), SERPINE1 (PAI-1), TFPI (TF pathway inhibitor), THBD (thrombomodulin), and vWF (von Willebrand factor), were not significantly different in ECs treated with serum from patients with non-COVID-19 or COVID-19 pneumonia. Although other pro-inflammatory cytokines and marker genes for endothelial activation (ICAM and PSELECTIN) were not evaluated in our study, the lack of activation of ECs after serum incubation, in this case, does not support the concept of endothelial damage caused by cytokines. Furthermore, it is possible that genes involved in endothelial PC signaling are not regulated on ECs, at least not while serum aPC levels protect the endothelium. As discussed above, endogenous serum aPC levels may not be constant throughout COVID-19. Therefore, the apparent protection observed here may not be sustained and are unable to avoid complications that compromise the patient’s life throughout the progression of the disease.

There are some limitations in our study. First, it was not assessed whether SARS-COV-2 would be capable of infecting other endothelial cell lines and then comparing the effects of infection with the effects of exposure to the virus alone. Second, it is important to note that our study did not analyze patient cohorts. Therefore, it is difficult to draw broad conclusions; yet we found interesting actions of the PC and aPC’s that deserve further investigation. Third, we used serum from patients infected with the initial strain of the virus. Therefore, we cannot rule out the possibility that currently prevalent or emergent variants could affect the endothelium in different ways.

Furthermore, we used HUVEC, a robust in vitro model, to investigate the biology of the human endothelium. However, ECs in districts such as the venous or arterial endothelium may behave differently. Finally, pharmacological treatments administered to patients with COVID-19, which may still be present in serum, could act as confounding factors, masking some effects on ECs. We performed in silico and cell culture experiments, and future in vivo experiments should test the effect of aPC.

## 5. Conclusions

The main finding of this study was that severe COVID-19 serum decreased the expression of inflammatory and pro-coagulation genes in HUVECs compared with healthy serum and culture medium. Interestingly, this decrease is associated with a higher serum level of aPC in patients with severe COVID-19 compared with healthy patients. It is known that under stress conditions, serum aPC levels naturally rise in response to tissue damage; however, these levels may subsequently fall, allowing the already-known correlation of initially high serum aPC levels being associated with a worse prognosis in COVID-19. Given the considerable number of patients with severe COVID-19 who die as a result of thrombo-inflammatory complications, even though initial serum levels of aPC can protect the EC from potential dysfunction, it is unlikely this proportion will be maintained to avoid significant endothelial damage. On the other hand, the inactive form of PC added before viral exposure showed significant modulation of *CCL2*, *IL6*, and *SERPINE1*, which can act as alternatives to protect the endothelial bed. However, the exact mechanisms by which PC or aPC carry out transcriptional modulation in HUVECs are unclear and should be investigated further.

## Figures and Tables

**Figure 1 viruses-16-01049-f001:**
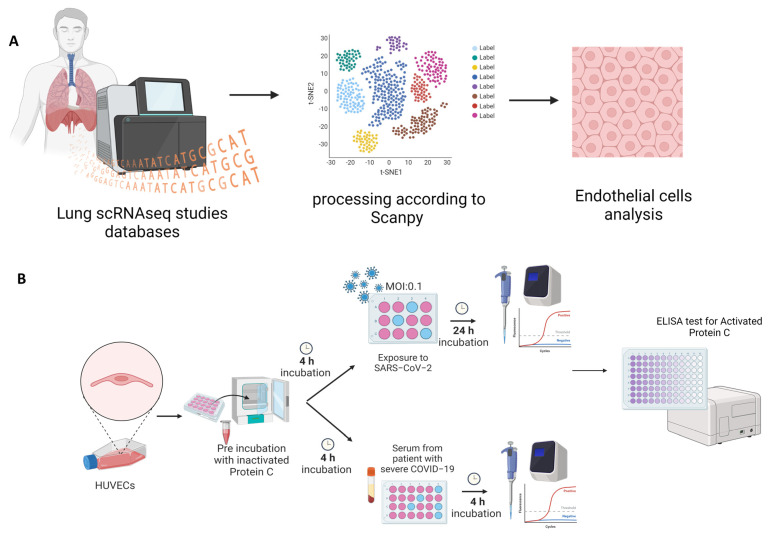
Overview of the study design. (**A**). Public data from single-cell RNA sequencing (scRNA-seq) studies on human pulmonary tissues were retrieved from the Gene Expression Omnibus (GEO) and European Genome-phenome Archive (EGA) under accession numbers GSE171668 and EGAS00001004344. (**B**). HUVEC of different passages pretreated or not with purified inactive human protein C for four hours at a concentration of 0.8 ng/μL. The upper panel shows HUVECs exposed to SARS-CoV-2 infection. Cells were grown in 12-well plates, with three wells per group—experimental N of 3. After incubation, the medium was collected for an ELISA test, and cell lysate was collected for subsequent RNA extraction and analysis by quantitative real-time PCR (RT-qPCR). The lower panel shows HUVECs incubated for 4 h with serum from patients with severe COVID-19. Cells were grown in 12-well plates, with two wells per group—experimental N of 4. After incubation, the medium was collected for an ELISA test, and the cell lysate was collected for subsequent RNA extraction and analysis by RT-qPCR.

**Figure 2 viruses-16-01049-f002:**
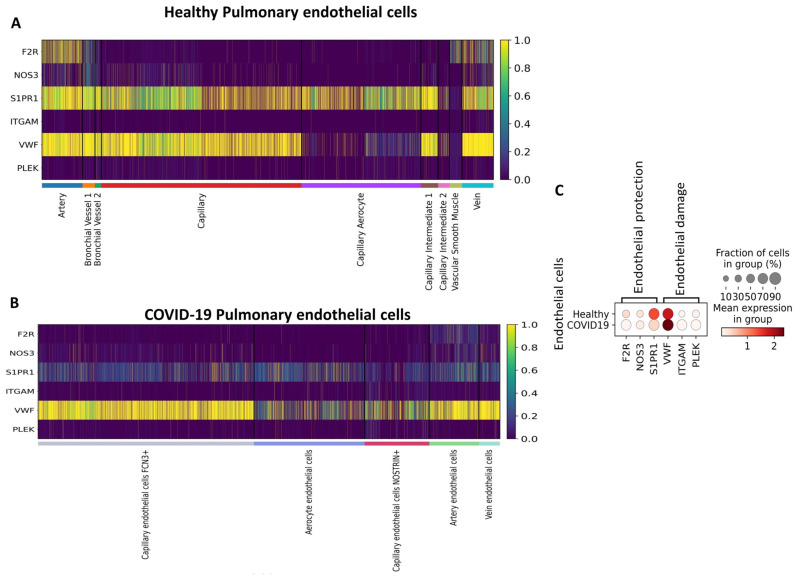
Gene expression of endothelial damage and the PC pathway in pulmonary endothelial cells of healthy patients and patients with coronavirus disease 2019 (COVID-19) based on single-cell transcriptomics. (**A**). Heatmap showing genes expressed among healthy pulmonary endothelial subtypes. (**B**). Heatmap showing genes expressed among COVID-19 pulmonary endothelial subtypes. (**C**). Dot plot representation of genes expressed in healthy *versus* COVID-19 pulmonary endothelial cells.

**Figure 3 viruses-16-01049-f003:**
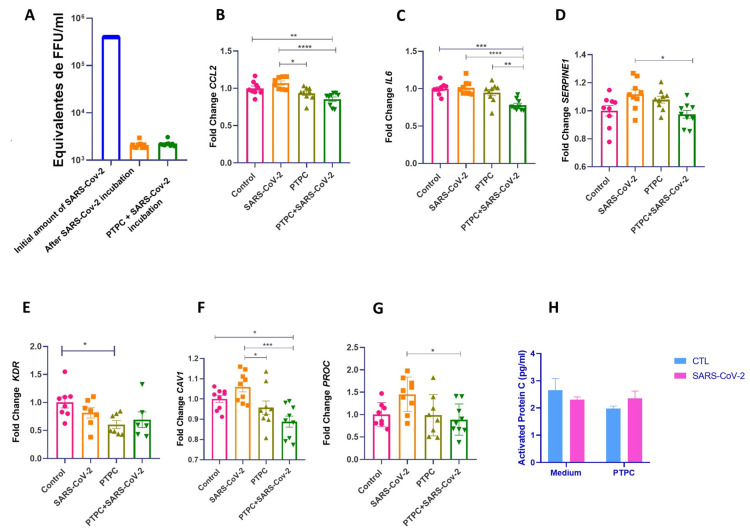
PC pretreatment reduces the expression of genes related to inflammation and coagulation in HUVECs exposed to SARS-CoV-2 infection. (**A**). Viral quantification of SARS-CoV-2 using real-time PCR on the medium collected after viral incubation. (**B**–**G**). HUVECs of different passages pretreated or not with purified inactive human protein C for 4 h at a concentration of 0.8 ng/μL and then exposed or not to SARS-CoV-2 infection. Cells were grown in 12-well plates, with three wells per group—experimental N of 3. The cells were washed with PBS before PC treatment, and a medium without FBS was used for the incubation. After the viral inoculum was removed, the medium was replaced with DMEM or MEM without FBS, and the cell lysate was collected for subsequent extraction and analysis of RNA by quantitative real-time PCR (RT-qPCR). One-way ANOVA followed by Tukey’s multiple comparison test was used for statistical analyses. * *p* ≤ 0.05, ** *p* < 0.01, *** *p* < 0.001, and **** *p* < 0.0001 compared with the groups indicated in the figure. (**H**). Activated protein C (aPC) levels were determined with an ELISA assay after incubation with SARS-CoV-2. The medium was collected after 24 h for the ELISA test. The two-way ANOVA followed by Tukey’s multiple-comparison test was used for statistical analyses. * *p* ≤ 0.05 compared with the indicated groups. Data are presented as mean ± s.e.m. PTPC: Pretreatment with PC. CTL: control—only cell culture medium was added.

**Figure 4 viruses-16-01049-f004:**
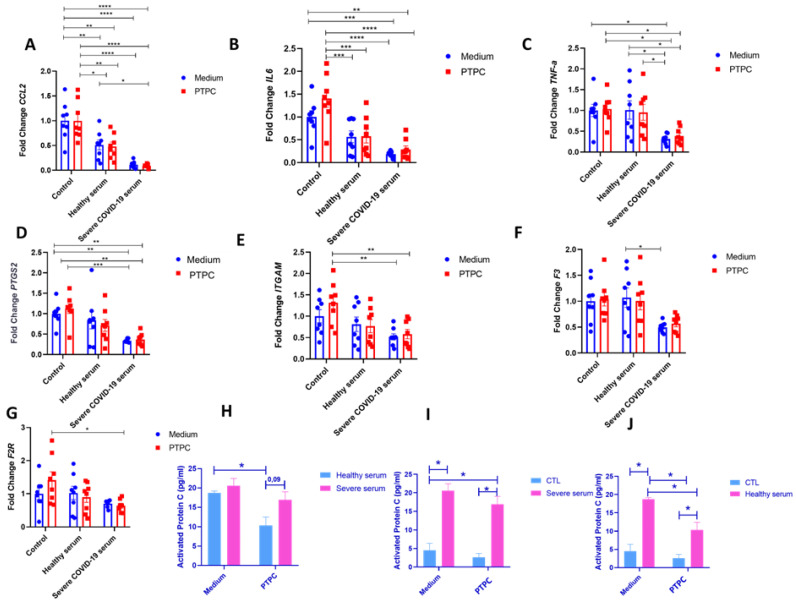
aPC present in severe COVID-19 serum may protect ECs from increased expression of inflammatory and pro-coagulation genes. (**A**–**G**). HUVECs of different passages pretreated or not with purified inactive human protein C for 4 h at a concentration of 0.8 ng/μL and then incubated with serum from a patient with severe COVID-19 or healthy serum at a concentration of 5%. Cells were grown in 12-well plates, with two wells per group—experimental N of 4. Cells were washed with PBS before PC treatment, and a medium without FBS was used for incubation. After serum incubation for 4 h, cell lysate was collected for subsequent extraction and analysis of RNA by quantitative real-time PCR (RT-qPCR). Two-way ANOVA followed by Tukey’s multiple comparison test was used for statistical analyses. * *p* ≤ 0.05, ** *p* < 0.01, *** *p* < 0.001 and **** *p* < 0.0001 compared with the groups indicated in the figure. Data are presented as mean  ±  s.e.m., with individual data points indicated. (**H**–**J**). Activated protein C (aPC) levels were determined after serum incubation with an ELISA assay. The medium was collected after 4 h for the ELISA test. Two-way ANOVA followed by Tukey’s multiple-comparison test was used for statistical analyses. * *p* ≤ 0.05 compared with the indicated groups. Data are presented as mean ± s.e.m. PTPC: Pretreatment with PC. CTL: control—only cell culture medium was added.

**Figure 5 viruses-16-01049-f005:**
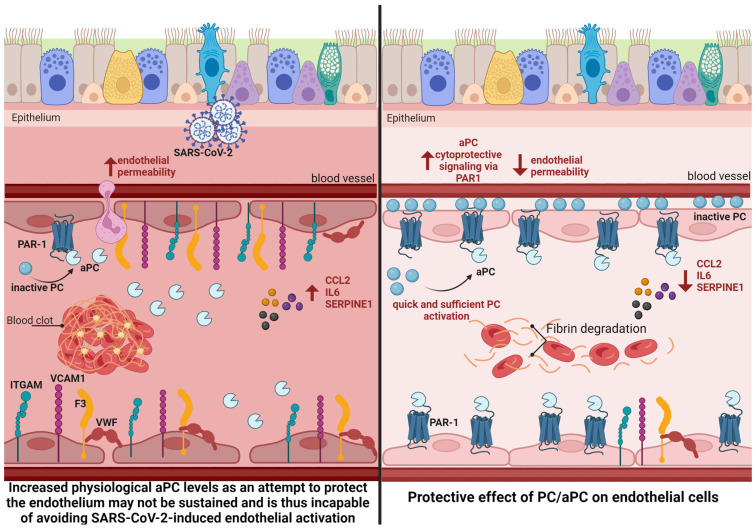
Panels comparing endothelial activation induced by SARS-CoV-2 and the cytoprotective effect of aPC. Left panel: The crucial role of endothelial cells in the pathogenesis of COVID-19 is related to the EC response to SARS-CoV-2 viral infection by detecting the adjacent infection and mounting a pro-inflammatory response to SARS-CoV-2 by stimulating circulating proteins. In addition, it is possible that even if direct infection in the ECs by SARS-CoV-2 does not occur, exposure to the virus is sufficient to promote transcriptional changes that lead to upregulation of pro-inflammatory and pro-coagulation genes and downregulation of cytoprotective signaling via Apc/PAR-1. Under stress conditions, serum aPC levels naturally rise in response to tissue damage; however, these levels may subsequently fall, allowing the already-known correlation of initially high serum aPC levels being associated with a worse prognosis in COVID-19. Right panel: The inactive PC might be stored in the EC (completing the PC reserve), and as soon as the cell is exposed to the virus, this protein promptly becomes activated and protects the vascular bed; consequently, its cytoprotective, anti-inflammatory, and anti-clotting signaling occurs more rapidly, thus actually protecting the cell from deleterious viral effects. PAR-1: Protease-activated receptor type 1; PC: Protein C; aPC: activated protein C; CCL2:C-C Motif Chemokine Ligand 2; ITGAM: Integrin Subunit Alpha M; VCAM1: vascular cell adhesion molecule 1; VWF: Von Willebrand factor.

**Table 1 viruses-16-01049-t001:** Differentially expressed genes related to protein C (PC) activation and signaling in lung endothelial cells from patients with coronavirus disease 2019 (COVID-19). Differentially expressed genes between COVID-19 patients and healthy controls were determined by implementing Scanpy’s two-sided Wilcoxon rank sum test. Genes without a value (indicated as ‘-’) did not reach the logFC threshold of 0.1. COVID19%: percentage of cells in the COVID-19 group expressing the gene. % Healthy: percentage of cells in the healthy group that express the gene. Abbreviations—logFC: log fold change.

		COVID-19 Lung Endothelial Cells			
Gene	Protein Product	LogFC	COVID19 %	Healthy %	Adj. *p*-Value
F2R	Coagulation factor II Thrombin receptor	−2.0546	71.63	36.43	3.3 × 10^−146^
ITGAM	Integrin Subunit Alpha M	2.0695	70.55	21.61	4.63 × 10^−22^
S1PR1	Sphingosine-1-Phosphate Receptor 1	−2.507	75.39	76.93	0
VWF	Von Willebrand Factor	0.8600	81.476	77.709	3 × 10^−133^
PLEK	Pleckstrin	2.2408	41.20	24.47	0.00073
NOS3	Nitric Oxide Synthase 3	−0.5492	43.73	27.46	6.62 × 10^−6^

## Data Availability

The data supporting reported results can be found in https://redu.unicamp.br/ (accessed on 19 June 2024).

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
