# Peer review of "Protein C Pretreatment Protects Endothelial Cells from SARS-CoV-2-Induced Activation"

_viruses, 2024, doi:10.3390/v16071049_

Round 1

Reviewer 1 Report

Comments and Suggestions for Authors

The work concerns the analysis of an experimental model in which HUVECs were treated with inactivated protein C before exposure to SARS-CoV-2 infection using An RT-qPCR array. Furthermore, active protein C levels were measured using the ELISA kit. The work seems well written and the results reported appear consistent with the investigations carried out. I would suggest the following changes:

- In the abstract I would modify the phrase “to explore the mechanisms behind the vascular and thrombotic events in COVID-19” as it appears too genetic

- In the introduction I would add more details on the mechanisms by which COVID-19 can cause coagulopathies, also referring to autopsy cases described in the literature

- I would insert a figure that summarizes the work carried out

Comments on the Quality of English Language

Good

Author Response

Dear Reviewer,
We appreciate your careful review of our manuscript. These changes will certainly greatly improve the quality of the article. Thank you very much.

  • In the abstract I would modify the phrase “to explore the mechanisms behind the vascular and thrombotic events in COVID-19” as it appears too genetic
  • The phrase was modified – line 17.

  • In the introduction I would add more details on the mechanisms by which COVID-19 can cause coagulopathies, also referring to autopsy cases described in the literature

  • In the introduction section more details on the mechanisms by which COVID-19 can cause coagulopathies were added. Lines 61-73.

  • I would insert a figure that summarizes the work carried out

  • The figure was added. Figure 5.

Reviewer 2 Report

Comments and Suggestions for Authors

Currently, there is no doubt that severe COVID-19 can cause pathological activation of the endothelium, leading to coagulopathy and microcirculatory disorders. The search for drugs to prevent these complications is an urgent problem of modern medicine. In their work, the authors found that the addition of PC to the medium reduces pro-inflammatory activity and protects endothelial cells from damage. The work is well organized, including the use of controls and comparison groups. The results have theoretical and practical relevance. However, I have a few comments:

(1) The authors did not consider clinical trial data showing the ineffectiveness of aPC in the treatment of patients with severe sepsis or septic shock [PMID: 22419295]. Therefore, the drug was withdrawn from the market after a subsequent placebo-controlled trial (Worldwide Evaluation of Severe Sepsis and Septic Shock with Human Recombinant Activated Protein C) [PMID: 24717456, this link may be omitted, provided for reference].

(2) Authors often use the term inactivated PC, but it is more correct to write inactive or non-activated, since inactivated usually refers to an irreversibly structurally altered molecule.

(3) In the Discussion section, the authors should consider recent data on the effect of SARS-CoV-2 virus on endothelial cells and other cells of the vascular network [PMID: 38576426].

Author Response

Dear Reviewer,
We appreciate your careful review of our manuscript. These changes will certainly greatly improve the quality of the article. Thank you very much!

(1) The authors did not consider clinical trial data showing the ineffectiveness of aPC in the treatment of patients with severe sepsis or septic shock [PMID: 22419295]. Therefore, the drug was withdrawn from the market after a subsequent placebo-controlled trial (Worldwide Evaluation of Severe Sepsis and Septic Shock with Human Recombinant Activated Protein C) [PMID: 24717456, this link may be omitted, provided for reference].

Clinical trials regarding aPC use in severe sepsis and septic shock were added in the introduction section. Lines 85-112.

(2) Authors often use the term inactivated PC, but it is more correct to write inactive or non-activated, since inactivated usually refers to an irreversibly structurally altered molecule.

All the terms inactivated were replaced by inactive. Lines 20,128, 147, 156, 157, 173, 176, 181, 285, 297, 305, 346, 577, 588.

(3) In the Discussion section, the authors should consider recent data on the effect of SARS-CoV-2 virus on endothelial cells and other cells of the vascular network [PMID: 38576426].

The exciting findings of the suggested study were included in the discussion section. Lines 381-386.

Round 2

Reviewer 2 Report

Comments and Suggestions for Authors

The authors have made the necessary changes to the content of the article. I have no further comments.